# Metal-Promoted Heterocyclization: A Heterosynthetic Approach to Face a Pandemic Crisis

**DOI:** 10.3390/molecules26092620

**Published:** 2021-04-29

**Authors:** Federico Vittorio Rossi, Dario Gentili, Enrico Marcantoni

**Affiliations:** 1Chemistry Division, School of Science and Technology, University of Camerino, 62032 Camerino, Italy; dario.gentili@unicam.it (D.G.); enrico.marcantoni@unicam.it (E.M.); 2Laboratori Alchemia Srl, Via San Faustino, 20134 Milano, Italy

**Keywords:** antiviral, heterocyclization, metal-promoted

## Abstract

The outbreak of SARS-CoV-2 has drastically changed our everyday life and the life of scientists from all over the world. In the last year, the scientific community has faced this worldwide threat using any tool available in order to find an effective response. The recent formulation, production, and ongoing administration of vaccines represent a starting point in the battle against SARS-CoV-2, but they cannot be the only aid available. In this regard, the use of drugs capable to mitigate and fight the virus is a crucial aspect of the pharmacological strategy. Among the plethora of approved drugs, a consistent element is a heterocyclic framework inside its skeleton. Heterocycles have played a pivotal role for decades in the pharmaceutical industry due to their high bioactivity derived from anticancer, antiviral, and anti-inflammatory capabilities. In this context, the development of new performing and sustainable synthetic strategies to obtain heterocyclic molecules has become a key focus of scientists. In this review, we present the recent trends in metal-promoted heterocyclization, and we focus our attention on the construction of heterocycles associated with the skeleton of drugs targeting SARS-CoV-2 coronavirus.

## 1. Introduction

The novel coronavirus disease (COVID-19) has resulted in a worldwide pandemic, with increasingly large numbers of infections [1]. Europe has recorded over 21.8 million infections and over 530,000 confirmed deaths [2]. The recent development of anti-COVID 19 vaccines has provided an effective weapon in the fight against this coronavirus [3,4,5]. Unfortunately, the persistent diffusion rate, the isolation of new variants, and the increasing mortality due to SARS-CoV-2 infection suggest that vaccines cannot be considered as the only clinical treatment [6]. Scientists from various fields have cooperated in the discovery and design of old drugs and potential new agents on both a small and a large scale [7]. However, promising strategies based on interferons [8] and monoclonal antibodies [9], as well as cell-based and immunopathology therapies, have resulted to be time- and cost-consuming and, thus, they cannot be considered valuable alternatives in the current conditions [10]. Researchers have focused their attention on current small molecules used as anti-influenza, antimalarial, and anti-HIV drugs. Various existing antiviral drugs have shown activity against COVID-19 both in vitro and in vivo, while some clinical trials have been conducted treating patients with these drugs or their combination [11,12,13]. Recently, pharmaceutical company Pfizer reported that their PF-0732 1332 molecule is currently in phase I clinical trials, representing the first orally administered clinical compound that targets the SARS-CoV-2-3CL protease. A common point among the different antiviral agents is the recurring presence of heterocyclic scaffolds (Figure 1). 

Heterocyclic compounds have versatile applications across many chemistry fields. N, S, and O are the most common heteroatoms, and their corresponding heterocycles can be found as the main structural units in synthetic pharmaceuticals and agrochemicals, as well as widely present in nature in plant alkaloids, nucleic acids, anthocyanins, and flavones [19]. Drugs containing a heterocyclic moiety inside their structure show antitumor, anti-inflammatory, antifungal, antidepressant, anti-HIV, antimalarial, and antiviral properties [20,21,22]. In particular, the latter three properties are central in the fight against SARS-CoV-2 [23,24,25]. Over the years, due to the importance of these small molecules, synthetic organic chemists have focused their efforts on the development of synthetic protocols which are more and more efficient, atom-economical, and environmentally friendly. Metal-catalyzed protocols, involving all metals from transition to rare-earth metals, have attracted the attention of chemists as compared to other synthetic methodologies because they directly employ easily available substrates to build multi-substituted complex molecules under mild conditions. Metal-catalyzed heterocyclization starting from acyclic precursors is considered a very performant tool in drug synthesis [26]. In this review, we focus our attention on metal-catalyzed heterocyclization methodologies for achieving pivotal scaffolds associated with molecules showing anti-COVID-19 properties. We describe recent applications in heterocycle synthesis, and we compare the classical synthetic routes and modern approaches used to obtain bioactive small molecules.

## 2. Chloroquine and Hydroxychloroquine

Chloroquine (CLQ) and its hydroxyl analogue hydroxychloroquine (CLQ-OH) were developed as antimalarial drugs, and they are used in the treatment of malaria, amebiasis, rheumatoid arthritis, and lupus erythematosus syndrome [10]. Both drugs show strong antiviral effects toward SARS-CoV-2 infection with calculated *IC_50_* values of 8.8 μM for CLQ and 5.47 μM for CLQ-OH [27,28]. Extensive clinical trials are ongoing to prove the efficacy of these drugs for treating COVID-19 infection [29]. They present a similar action mechanism; chloroquine and hydroxychloroquine are able to modify the pH of host cell lysosomes. This pH increase corresponds to a modification of the cellular biological activity, leading to a cascade of processes which prevent cellular replication [30]. The fundamental effect of chloroquine and hydroxychloroquine in the treatment of different pathologies has spurred chemists to establish various routes for their synthesis. Figure 2 reports the key intermediates used in the main strategies developed over the decades.

Synthetic routes for chloroquine are based on harsh conditions that promote byproduct formation and low overall yield of the whole process. In the first known synthesis of chloroquine, reported by Surrey and Hammer, formation of the pivotal quinoline core **2** was carried out at high temperature, which promoted the formation of undesirable isomers **2′** and **3′**. Moreover, the decarboxylation step, promoted by a strong base and a mineral acid, is not considered sustainable (Scheme 1) [31].

Jonnson and Buell later developed a CLQ synthesis method with an improved overall yield of 25%. Unfortunately, the formation of the quinoline moiety led to the easy formation of byproducts due to the strong reaction conditions (Scheme 2) [32].

In 2007, Margolis et al. proposed a synthetic route to achieve CLQ. The relatively mild conditions of the process made it suitable for large-scale production; however, in this case, the formation of the quinoline scaffold was also promoted at high temperature, thereby favoring byproduct formation (Scheme 3) [33].

Even the synthetic methodologies developed for hydroxychloroquine feature critical steps. Hammer and coworkers, in their three-step, synthesis proposed obtaining the target via an S_N_Ar between intermediate **8** and dichloroquinoline **3** as the final step. Low overall yield, use of phenol as the solvent, and high reaction temperature hindered the scale-up of this strategy (Scheme 4) [34].

Kumar and coworkers, inspired by Hammer’s work, modified the synthetic protocol and enhanced the overall yield from 18% to 40%. However, the final S_N_Ar step to achieve CLQ-OH was carried out in harsh conditions (high temperature and long reaction time) (Scheme 5) [35].

Recently, Min et al. proposed an alternative approach to functionalize quinoline **3**; however, the use of high pressure in combination with high temperature represents a safety concern (Scheme 6) [36].

Yu and Gupton exploited the continuous-flow methodology to improve the process from an industrial point of view. Starting from 2-acetylcyclopentan-1-one **9**, they were able to synthesize the key intermediate **8** while achieving a yield improvement of 52% compared to classical processes. Unfortunately, even in this case, the C–N coupling to access hydroxychloroquine was carried out in unsustainable conditions (Scheme 7) [37].

### Quinoline Synthesis: Metal-Promoted Annulation

The biological importance of quinoline-based drugs has resulted in the synthesis of this substituted heterocycle becoming a hot topic for organic chemists worldwide [38,39,40,41]. A plethora of elegant syntheses have been developed; however, the use of harsh conditions and limitations due to the nature of some reagents have restricted the application of these protocols both in academia and in industry [42]. The recent trend of obtaining targets with high purity using sustainable conditions has resulted in the use of metal catalysts becoming central in the synthetic strategies of complex drugs.

Friedländer synthesis using 2-aminobenzaldehyde and carbonyl derivatives has been exploited for a long time to obtain substituted quinolines. Currently, modifications of this methodology have permitted the development of efficient and elegant protocols for the synthesis of this heterocyclic framework (Figure 3).

Among the different metal catalysts exploited to trigger Friedländer condensation, Ru(II) complexes were found to be very effective. Their relative cheapness and safety in handling make them good promoting systems to perform scale-up.

Yus et al. studied the condensation between (2-aminophenyl)(phenyl)methanol **10** and ketones **11** for the formation of 2,3,4-substituted quinolines **12**. The reaction is promoted by RuCl_2_(DMSO)_4_, and its ability to accept and donate H_2_, thereby restoring its original oxidation, is crucial for the catalytic cycle (Scheme 8) [43].

Optimized reaction conditions permit obtaining polysubstituted quinolines at sufficient to excellent yields in relatively mild conditions (e.g., **12 a–d**), producing water as waste. The addition of benzophenone acting as a hydrogen scavenger allows improving the final yield of the targets. This result can be explained by the partial inability of ruthenium hydride species to restore the catalytic cycle (Scheme 9) [44].

The same synthetic protocol was applied to both sterically hindered ketones **11** and various anilines **13** for the formation of the desired quinoline derivatives **14 a–c** (Scheme 10).

Yus proved the versatility of RuCl_2_(DMSO)_4_ as a catalyst in the hydrogen-borrowing process to obtain substituted quinolines **12, 14** by exploiting the reactivity of secondary alcohols **15** with (2-aminophenyl)methanol **10** (Scheme 11) [43,45].

The plausible catalytic cycle involves the formation of the active corresponding potassium alkoxides. The subsequent oxidation/condensation cascade leads to the formation of the target quinoline (Scheme 12) [43].

In addition to RuCl_2_(DMSO)_4_, an indirect Friedländer process was reportedly promoted by iridium, palladium, copper, and rhodium complexes [46,47,48,49,50,51]. Figure 4 presents the common catalysts used in the annulation between aniline derivatives and hydroxylic scaffolds to achieve quinoline motifs.

An effective alternative to the indirect Friedländer approach is represented by the one-pot alkynylation/cyclization protocol using aniline derivatives, substituted alkynes, and aldehydes. In 2016, Maiti et al. proposed an innovative solvent-free CuBr–ZnI_2_ catalytic strategy to afford polysubstituted quinolines and chiral sugar-based quinolines (**19 a–d**) in sufficient to good yields (Scheme 13) [52].

In this three-component protocol, substituted aniline **16**, terminal alkynes **17**, and aldehydes **18** react fast and in mild conditions through C–C and C–N bond formation promoted by Zn(II) and an C(*sp^2^*)–H activation promoted by Cu (I) and the transient formation of aryl Cu(III) species, followed by subsequent cyclization.

A comparable protocol was developed by Sarode and coworkers. They showed the catalytic ability of zinc(II) triflate to promote multicomponent C–C and C–N formation using anilines **16**, terminal alkynes **17**, and aryl aldehydes **20** in solvent-free conditions (Scheme 14) [53]. 

The use of inexpensive catalysts, the absence of toxic solvents and additives, and the tolerance toward different functional groups make this reaction a great candidate for scale-up processes.

Korivi and Cheng exploited Ni catalysis to assist the annulation between iodo-anilines **21** and aroylakynes **22** (Scheme 15) [54].

This methodology permits achieving a broad range of 2,4-disubstitued quinolines **19** in satisfactory yields. The Ni catalyst does not need an extreme inert atmosphere to work. Zn powder is necessary to regenerate the initial oxidation state of the nickel catalyst from Ni(II) to Ni(0).

Recently, aroylakynes were exploited by Liu and coworkers to access the complex quinoline scaffold **24** (Scheme 16) [55].

In this procedure, the catalytic system (Ph_3_P)AuCl/AgOTf promotes the cycloaddition between 2-aminoaryl carbonyls **23** and internal alkynes **22** at good to excellent yields (e.g., **24 a–d**) in sustainable conditions, affording a plethora of polysubstituted quinolines **24** containing various functional groups. The presence of Ag(I) salt as an additive was crucial for the activation of the catalyst due to the ability of silver to dechlorinate the Au catalyst, thereby increasing the electrophilicity of the metal center. The procedure exhibits adaptability to different functional groups using both internal alkynes and aminoaryl derivatives, leading to a wide array of substrates.

The efficiency of gold catalysis was shown in the work of Ji et al. The same promoting system displayed high efficiency in the cyclization of 2-trifluoromethylated propargylamines **25** (Scheme 17) [56].

A gold(I) catalyst triggers the internal cyclization of propargylamines to obtain diverse quinolines **26**. Mild conditions and a broad scope of the reaction were attained using this methodology. It is important to highlight the facile introduction of a fluorinated moiety into the target, considering the biological significance of fluorinated quinolines.

An innovative and elegant pathway to achieve polysubstituted quinolines **29** was proposed by Xu et al., whereby an Ag(I) catalyst promotes 6-endo-dig cyclization of 2-azide alkyne derivatives **27** followed by an R–X **28** insertion into the imino carbene generated in the catalytic cycle (Scheme 18) [57].

Readily available materials, the cheap silver catalyst, and mild reaction conditions make this procedure appealing for organic chemists. The introduction of halogens into the heterocyclic scaffold provides the possibility of target derivatization to access various quinolines.

## 3. Arbidol

Arbidol (uminefovir) is an oral antiviral drug with a broad spectrum of activity against many types of viruses. It has been licensed for the treatment of influenza A and B in Russia since 2003 and in China since 2006 [58]. Arbidol is a non-nucleoside membrane fusion inhibitor that prevents the interaction of the influenza virus with the host cell. Arbidol shows a binding mode with the SARS-CoV-2 spike protein similar to that with influenza virus hemagglutinin (HA) [59,60]. SAR studies on Arbidol have indicated that the indole core and the thiophenyl motifs are pivotal for the molecule bioactivity (Figure 5). 

The lipophilicity of the previous synthons permits them to penetrate the hydrophobic cavity of influenza virus HA, thereby inducing a structural change and the consequential break of the salt bridge between the virus and hosting cell [61]. The identification of potential scaffolds inside the drug has permitted scientists to apply a synthetic chemistry approach to their modification them in order to enhance their antiviral activity and synthesize analogues with increased anti-COVID 19 properties [62].

The first synthetic approach to obtain Arbidol was reported in 1993 by Trofimov, involving decoration of the aromatic ring of the indole derivatives **30** previously synthesized by the same group (Scheme 19) [63]. 

This approach has been widely employed in large-scale production despite the use of toxic solvents such as carbon tetrachloride or formaldehyde and dimethylamine for the final Mannich reaction. In the last year, the total synthesis of this bioactive compound was renewed beginning from the synthesis of the crucial intermediate **30**.

Gong and coworkers described the synthesis of various ethyl 5-hydroxy-1*H*-indole-3-carboxylates **37** with anti-hepatitis B activity. To achieve the target compounds, formation of the intermediate **36** was used as a precursor of Arbidol starting from commercially available ethyl 4-chloro-3-oxobutanoate **33** (Scheme 20) [64].

A Nenitzescu reaction between enamine **35** and 1,4-benzoquinone permits accessing the 5-hydroxyindole intermediate **36** in sufficient yield and with punctual regiochemistry. Successive decorations of the indole ring lead to the construction of antiviral target compounds **37**. 

In the last decade, the ongoing interest around Arbidol due to its antiviral properties has led to it becoming a target for API producers. In 2016, Gao et al. developed a total synthesis protocol for Arbidol starting from nitrophenol **38** (Scheme 21) [65].

This eight-step process leads to the formation of Arbidol in excellent overall yield. The enamine **43** is pivotal for the construction of the Arbidol indole scaffold. In this work, enamine oxidative cyclization triggered by Pd(II) was exploited to achieve substituted indole in mild conditions while minimizing regioisomeric drawbacks. Unfortunately, the use of a large excess of Cu(OAc)_2_ as an oxidant is necessary to restore Pd(0) to Pd(II), which makes this cyclization less attractive from a synthetic point of view.

Its recent commercialization, the establishment of various synthetic protocols, and its use as a potential candidate in the therapy against SARS-Cov-2 have enhanced the interest in Arbidol. The indole scaffold has emerged as central in the existing synthesis protocols; thus, the development of alternative indole synthesis approaches involving different starting materials and metal catalysts may lead to accelerated production of this API.

### Metal-Promoted Heterocyclization to Achieve Polysubstituted Indoles

Indole is one of the most common heterocyclic scaffolds, used in a large array of drugs, natural products, and agrochemicals. The importance of this aromatic *N*-heterocycle has been highlighted by the continuous work carried out on it [66]. In this section, we suggest some recent metal-catalyzed heterocyclization pathways to achieve polysubstituted indoles in an easy and accessible way with the aim of finding plausible alternative strategies for the synthesis of the indole core present in Arbidol.

Ruchirawat et al. came up with an efficient and easy procedure for accessing a plethora of substituted indoles **45** (Scheme 22) [67].

Stable and easily accessible 2-alkynylanilines **44** are employed in this intramolecular cyclization catalyzed by PtCl_4_ in mild conditions with a high tolerance toward various functional groups. The catalyst was found to be performant with low loading (1–2 mol.%) and it did not need an elevated reaction temperature. Most substrates are converted to the desired target at room temperature in high yields (e.g., **45 a,b,d**), whereas refluxing conditions are needed with halogens or EWG groups in the 7-position; however, in this case, excellent conversion is also afforded (e.g., **45 c**).

Recently, Co(III) catalysts have attracted the interest of chemists due to their versatility and selectivity in various chemical transformations. Liang and Jiao, in their work, proposed an interesting indole synthesis approach triggered by an inexpensive cationic cobalt(III) complex starting from readily accessible *N*-nitrosoanilines **46** and substituted alkynes **47** (Scheme 23) [68].

Co(III) catalysts have proven to be suitable for the efficient and direct cyclization of nitroso derivatives with internal alkynes. This methodology provides a large array of polysubstituted indoles bearing different functionalities ranging from halogens to electron-withdrawing and alkyl groups (e.g., **48 a–d**). An improved regioselectivity, low-cost rare abundant metal catalyst, and the lack of strong oxidizing agents make this protocol attractive for both academia and industry.

The flexibility of Cp*Co(III) was also demonstrated by Glorious and coworkers. In their work, they proposed a switchable cyclization between *N-*phenylalkylamides **49** and alkynes **47** to obtain quinolines and indoles (Scheme 24) [69].

The selectivity on this protocol is based on the reactivity of an organometallic intermediate which can undergo dehydrative cyclization copromoted by Lewis acids to form substituted quinolines or dehydrogenative cyclization to form indoles. This methodology leads to decorated and synthetically precious scaffolds; however, it necessitates a high reaction temperature and the use of a stoichiometric oxidizing agent to promote the cyclization of starting materials.

Among the rare abundant metals, manganese complexes also show the ability to trigger the formation of indole scaffolds. Wan’s research group proposed an unprecedented coupling reaction between aromatic amines **51** and diazo compounds **52**. This Mn(II)-catalyzed tandem reaction provided a different and practical approach for obtaining an indole skeleton under mild conditions (Scheme 25) [70].

This novel methodology based on a radical–carbene coupling reaction permits obtaining a plethora of interesting indoles bearing various functionalities. Mild conditions, readily available or easy-to-prepare precursors, and the cheap and easy-to-handle Mn(OAc)_2_·4H_2_O catalyst make this synthetic procedure attractive for affording *N*-heterocycles. 

As previously shown in the Arbidol synthetic approach, the construction of 5-hydroxy indole derivatives is pivotal due to their presence in molecules showing biological activity (e.g., serotonin). Unsworth and coworkers presented an efficient silver(I)-catalyzed “back-to-front” cyclization of alkyne in a pyrrole scaffold (Scheme 26) [71].

The proposed methodology presents a unique reaction mechanism. DFT studies on pyrrole-ynones **55** have shown that the activation of the alkyne moiety via silver transmetalation is followed by an unusual pyrrole C-3 position nucleophilic attack against the formal C-2 position. Additionally, the reactions proceed at room temperature in sustainable conditions, affording indole targets in excellent yields.

Pd(II) complexes can be used to efficiently synthesize substituted indole scaffolds. Recently, Youn and coworkers in their work described Pd-catalyzed annulative couplings of 2-alkenylanilines **59** with aldehydes **60** using alcohols both as a solvent and as a hydrogen source (Scheme 27) [72].

Youn demonstrated high regioselectivity related to a hydropalladation process and exploited the reactivity of an alkylpalladium intermediate with imines. The use of alcohol both as a solvent and as a hydrogen donor represent for the uniqueness of this reaction. A large scope of methodology and good yields are achieved, and the process has even been applied in the synthesis of compound **61 d**, a calorimetric detector of DNA [73]. Unfortunately, the high reaction temperature (120 °C) and air-sensitive catalyst remain major barriers to the scale-up of this methodology.

Among the different synthetic processes developed over the years, metal-catalyzed dehydrogenative annulation has shown the selectivity of metal-catalyzed reactions along with enhancing process sustainability. Usually, these kinds of reactions proceed in mild conditions, exploiting green and renewable sources such as alcohols or glycols with the formation of water or ammonia as principal byproducts. These cyclizations can be triggered by a plethora of metal catalysts, leading to multi-decorated indole frameworks (Scheme 28) [74].

## 4. Telmisartan

Telmisartan (commercial name Micardis^®^) is a potent and selective angiotensin II type 1 (AT_1_) receptor antagonist. It is characterized by excellent AT_1_ receptor-binding activity, a long half-life, and good tolerability (Figure 6) [75]. 

Recently, it was demonstrated that the SARS-CoV-2 virus binds to the membrane protein ACE2 (angiotensin-converting enzyme 2) of cells via its S protein (spike). ACE2 catalyzes the transformation of angiotensin II (apoptosis and inflammatory effects) to angiotensin 1–7 (anti-inflammatory effects) [77]. The presence of the virus, thus, reduces the formation of a natural antagonist of angiotensin II. According to this plausible mechanism, ongoing pharmaceutical therapies are based on the administration of drugs decreasing the activity of angiotensin II, e.g., classical ACE inhibitors or angiotensin II type 1 (AT_1_) receptor antagonists such as telmisartan [78]. Over the decades, various synthetic approaches have been developed to obtain the *bis*-benzimidazolic framework of telmisartan. The first reported synthesis of this drug was presented by Ries and coworkers in 1993 (Scheme 29) [79].

This eight-step process faces some synthetic issues. The nitrosation of **62** is based on the use of a large amount of strong mineral acids, which represents both a safety and a wastewater disposal concern. Moreover, the use of polyphosphoric acid (PPA) to promote cyclization to achieve intermediate **65** is associated with tricky operation conditions and work-up. The alkylation of **65** with **66** leads to regioisomer formation, and this corresponds to an increase in the difficulty of the purification of the final API and a lower overall yield (21% yield in eight steps).

Recently, the growing interest around telmisartan has spurred synthetic chemists to improve the known synthetic strategy or to find alternative and more performant pathways to obtain this drug.

In 2020, Shen et al. designed an efficient synthetic route for telmisartan. They focused their attention on the synthesis of the bis-benzimidazole intermediate **65** via Cu catalysis, avoiding PPA as a condensing agent (Scheme 30) [80].

The improved methodology is based on the ability of the Cu(I) salt to promote the annulation of benzimidazolyl-substitued *o*-haloyralamidines **68** and **69** to their corresponding *bis*-benzimidazolic scaffolds. This seven-step process achieve the desired API in 54% overall yield while enhancing the safety, operability, and sustainability of the original protocol. The use of DMSO constitutes a process issue because the solvent is unrecoverable. 

Xiang’s research group described the use of a green inorganic salt to promote the synthesis of the benzimidazolic framework. They exploited Na_2_S_2_O_4_ in a protic solvent to obtain the key intermediate **65** in an excellent 85% yield (Scheme 31) [81].

Starting from simple and commercially available starting materials, Xiang accomplished a seven-step total synthesis of telmisartan in good yield with a really high purity of the final API (99.7% HPLC). The reductive cyclization to obtain **65** employs inexpensive sodium dithionite (Na_2_S_2_O_4_), which acts as a single electron donor to reduce the nitro group in **71** to an amino group and promotes condensation/cyclization with butanal **72**. An important drawback of this reaction is linked to the high loading of reducing agent (600 mol.%) needed to promote the reaction with high conversion to the desired target.

Gupton et al. converted the batch approach in flow-based synthesis to obtain the target drug. The convergent process exploited a Suzuki cross-coupling between **77** and **78** to afford telmisartan in 81% yield over three steps (Scheme 32) [82].

This approach required no intermediate isolation or solvent exchange, implying an improvement over existing batch methods that need numerous additional operations that add complexity and waste to the overall process.

### Metal-Catalyzed Annulation in Benzimidazole Synthesis

The importance of benzimidazole and its derivatives, such as *bis-*benzimidazoles, due their broad biological activities, makes them desirable scaffolds for the pharmaceutical industry [83]. The increasing demand for drugs featuring a benzimidazole framework has led to organic chemists seeking alternative pathways or renewing classical methodologies to obtain this *N*-heterocycle. Typical synthetic procedures involve the use of a stoichiometric amount of strong acids, high reaction temperatures or reaction auxiliaries leading to undesired byproducts, and a lot of waste formation [84]. The synthesis of benzimidazoles should accomplish process sustainability by using renewable resources with high atom economy. In this section, we present recently proposed sustainable metal-catalyzed methodologies to achieve substituted benzimidazoles.

Milstein et al. suggested a cobalt-catalyzed dehydrogenative coupling between easily accessible aryl diamines **79** and alcohols **80** to obtain 2-substituted benzimidazoles with high selectivity (Scheme 33) [85].

This appealing methodology permits achieving 2-substituted benzimidazoles in outstanding yields (e.g., **81 a–d**), avoiding the use of additives or an exogenous base. The hydride source NaBEt_3_H is necessary to activate the catalyst reduction from Co(II) to Co(I). The use of an earth abundant metal catalyst and the formation of water and hydrogen as reaction byproducts make this methodology highly sustainable and applicable in pharmaceutical synthesis.

A similar dehydrogenative approach to obtain benzimidazole was adapted by Hong and coworkers. They tested, for the first time, the ability of a Knölker-type catalyst, tricarbonyl (η^4^-cyclopentadione) iron complexes, to promote oxidative coupling between substituted aryl diamines and alcohols to produce *N-*alkylated benzimidazoles (Scheme 34) [86].

The first reported case of iron-catalyzed dehydrogenative coupling allowed obtaining substituted benzimidazoles in great yields (e.g*.,*
**82 a–d**) in relatively mild conditions. The regioselectivity of the reaction is modifiable as a function of the electron density of the group tethered to alcohols (electron-rich heteroaromatic substituents give reverse regioselectivity). The main limitation is represented by the excess use of *t*-BuOK to promote the annulation process. To avoid initial alcohol oxidation, the CeCl_3_–CuI catalytic system promotes the one-pot cyclo-dehydrogenation of aniline Schiff bases generated “in situ” from the condensation of **79** and aldehydes. Subsequent oxidation afforded benzimidazoquinazoline derivatives that present antiviral activity [87].

The use of cheap and easy-to-handle Cu(I) to catalyze various reactions has increased over the decades. Zhou’s research group tested Cu(I)’s ability to promote benzimidazole formation in a three-component one-pot process starting from 2-haloaniline **83**, arylic aldehydes **84**, and ammonia (Scheme 35) [88].

The large reaction scope, the inexpensive but performant Cu(I)–Cu(III)catalytic system, the high tolerability toward various functional groups, and the use of water as a solvent make this synthetic procedure attractive for the future synthesis of complex bioactive compounds with a benzimidazole scaffold.

The ability of Cu salts to promote *N*-heterocycle formation was highlighted in Zhang’s work. Synthesis of 5-diarylamino benzimidazoles **88** was accomplished via a radical-induced tandem triple C–H activation starting from secondary aromatic amines **87** and aminating agents **86** (Scheme 36) [89]. 

The unprecedented aerobic copper-catalyzed synthesis of benzimidazoles **88** proceeds with high conversion, selectivity of desired targets, and excellent tolerability toward a broad array of functional groups. The low-cost catalytic system, readily available reagents, and atom efficiency make this process intriguing and desirable for benzimidazole construction.

Li et al. obtained substituted benzimidazoles **91** by exploiting Ir(III) catalysis under redox neutral conditions, triggering a C–H activation–amidation–cyclization pathway (Scheme 37) [90].

Ir(III) annulative coupling between *N*-functionalized anilines **89** and dioxazolones **90** led to the desired target’s formation in sufficient to good yields (e.g., **91 a**–**d**). High chemo- and regioselectivity were achieved. The only reaction byproducts were CO_2_ and H_2_O, providing excellent atom economy.

## 5. Quercetin and Luteolin

Quercetin and Luteolin are natural products derived from plants, known as phytochemicals, and they belong to the flavonoid category [91].

This class of compounds is hardly toxic and presents considerable antiviral properties. The bioactive properties of flavonoids were demonstrated by the large number of antiviral medications produced between 1981 and 2006 [92]. Recently, flavonoids have been extensively studied for targeting the spike protein of SARS-CoV-2 to restrict virus access to host cells and for inhibiting the SARS-CoV 3CL protease to decrease viral infectivity. Molecular docking software identified quercetin and luteolin to be the best candidates for COVID-19 inhibitors [93,94,95,96].

Flavonoids are biosynthesized by plants starting from phenylalanine, which is rapidly converted to 4-coumaroyl-CoA. Malonyl CoA reacts in a 3:1 ratio with the coumayl-CoA derivative to give the key intermediate naringenin, catalyzed by chalchone synthase. Two different pathways lead to the formation of quercetin (via hydroxylation, promoted by flavone 3-hydroxylase F3H and dehydrogenation) and luteolin (via dehydration, promoted by flavone synthetase SI) (Scheme 38) [97].

Promising preliminary studies both in vitro and in vivo, readily available cheap sources, and the lack of severe side-effects have led to increased interest in flavonoids as potential therapeutic agents against COVID-19 infections [98]. The flavonoid approach to treat respiratory infections due to COVID-19 has emerged as a valid pharmacological alternative, but some aspects remain questionable. Some flavonoid plant sources such as peanut shells or fava beans contain allergenic substances that can affect patients; moreover, the poor oral absorption of flavonoids makes their administration a tricky process [99]. To overcome the problems related to the use of these natural compounds, alternative synthetic routes and decoration pathways are essential. Recently, flavonoid synthesis has been studied intensively by organic synthetic chemists, and a plethora of metal-catalyzed synthesis approaches have been proposed to enlarge the scope of these compounds. In the next paragraph, we report interesting examples of nonbiomimetic flavonoid synthesis starting from available and stable materials. 

### Metal-Catalyzed O-Heterocyclization to Flavonoids

The flavonoid framework is recurrent in drugs and natural products, showing unique biological properties and physiological actions. Due to their varied biomedical applications, flavones have aroused great interest in the chemistry community, leading to the development of performant and sustainable synthesis and functionalization approaches in the last decade. Metal-catalyzed heterocyclization represents an outstanding and selective strategy to obtain these scaffolds starting from readily available or easy-to-synthesize starting materials. Below, recent strategies are reported for the synthesis of substituted flavones.

Liu et al., in their work, proposed the palladium-catalyzed dehydrogenative annulation of *o*-acyl phenols **92** to flavones **93** (Scheme 39) [100].

Pd(0) exhibits high activity in promoting the C–H functionalization of electron-rich *o*-acyl phenols, followed by C–O bond formation and relative annulation. The reaction proceeds without oxidants and hydrogen acceptors which can lead to byproduct formation. The high isolated yield of flavonoid derivatives, low catalyst loading, recoverable heterogeneous catalytic system, and broad substrate scope make this methodology appealing for industrial purposes.

In 2017, Lee’s research group exploited [Ru(*p*-cymene)_2_Cl_2_]_2_ to promote the C–H activation of salicylaldehyde **94** and trigger decarboxylative coupling with alkynoic acids **95** to reach obtain flavonoids **96, 97** (Scheme 40) [101].

This simple and selective one-pot metal-catalyzed synthetic methodology leads to the formation of a large array of homoisoflavonoids **97** and flavones **96** in good yields, starting from cheap salicylaldehyde and easily accessible propiolic acid. The selectivity of the two targets is determined by the solvent choice: in *t*-amyl alcohol, **96** is dominant, whereas the use of DMSO produces **97**.

The potential of a metal-promoted decarboxylative coupling/annulation protocol was exploited by Qi and coworkers. They utilized Ag(I) salt to trigger radical tandem alkynylation/annulation to yield a plethora of flavones **99** in mild conditions, identifying arylpropiolic acids **95** and α-keto acids **98** as valid precursors for the reaction (Scheme 41) [102].

This new mild Ag(I) radical approach was found to be performant for obtaining flavonoid frameworks in sufficient to good yields and with high tolerability toward the functional groups present in the scaffolds. Persulfate is necessary to re-oxidize Ag(I) to the active species Ag(II).

Among the different transition metals, iridium complexes have emerged as powerful catalysts to promote C–O and C–C bond formation in a highly selective fashion. 

Wu et al. tested the ability of iridium complexes to obtain flavones. Simple phenols, internal symmetric alkynes, and gaseous CO were employed as starting materials; the ligand choice in the catalytic system is crucial for the success of transformation (Scheme 42) [103].

This novel carbonylative cyclization permits producing 2,3-disubstitued flavone scaffolds in very good isolated yields. The carbonylation is effective on non-preactivated phenols and alkynes. Limitations of the methodology are related to the use of a high amount of oxidizing agent and high-pressure CO, which represents a safety concern due its toxicity.

The use of Lewis acids has emerged as a potent tool to promote heterocyclization and obtain a plethora of different heterocyclic scaffolds. Among Lewis acids, the use of metal halides is related to mild reaction conditions and sustainability of the process, making them desirable as promoting agents or catalysts [104].

Van Lier and coworkers developed silica gel-supported indium halides as a heterogeneous catalytic system to access flavones **103** from corresponding 2′-hydroxychalcones **102** (Scheme 43) [105].

The rapid intramolecular oxidative coupling of **102** to flavones was accomplished in elevated yields. The sustainability of this methodology is fulfilled by the use of inexpensive, commercially available indium salts in neat conditions while avoiding the formation of byproducts.

Lewis acid catalysis in flavone synthesis was demonstrated in Su’s work. Di-carbonyl compounds **104** were annulated efficiently in the correspondent flavones **105** in the presence of Ga(OTf)_3_ as a mild promoter (Scheme 44) [106]. 

The reaction scope encompasses a wide array of polysubstituted targets in high yields. The protocol is operationally simple thanks to a short reaction time, straightforward work-up, and easy metal triflate recoverability.

## 6. SARS-CoV-2 3CL Protease Target Drugs

The SARS-CoV-2 3C-like protease is the main protease present in the virus, and it is crucial in the translation process from polyproteins to viral RNA [107]. It was demonstrated that the catalytic domain (Cys-145 and His-41) is particularly conserved, which makes the 3CL protease an attractive target for broad-spectrum anti-coronavirus therapies and drug discovery [108]. The SARS-CoV-2 main protease and spike protein are essential for the transmission of the virus and the severity of the infection in the host. Suppressing one or both biological targets can address the concerns linked to transmission, whereby acute COVID-19 symptoms can be drastically minimized [109]. Potential 3CL protease inhibitors reported in the literature have been screened to test their efficacy. Among the prospective bioactive molecules targeting this protein, ritonavir in combination with lopinavir and *N*-decorated isatins has shown promising results in the fight against SARS-CoV-2 [16,110,111]. 

Ritonavir (branded name (Norvir^®^) is a powerful human immunodeficiency virus (HIV) protease inhibitor with effective antiretroviral activity. The use of a therapeutic dose of the drug (600 mg per day) has been linked to gastrointestinal and neurological toxicity as side-effects. However, subsequent pharmacokinetic studies have shown the efficacy of ritonavir, in low dosage, as a drug “booster” [112]. Coadministration with different protease inhibitors coincides with their increased concentration in plasma, their increased elimination half-life, and reduced food influence on their gastrointestinal absorption [113]. The elongated shape and the presence of a thiazole scaffold inside the drug is pivotal for the boosted activity. It acts as a CYP3A4 inhibitor, attaching irreversibly to the heme or amino-acid residues via the thiazole interaction [114]. Recent in silico simulations showed the binding ability and selectivity of the lopinavir/ritonavir drug combination toward the SARS-CoV-2 3CL protease. These results confirm the utility of anti-HIV drugs in the fight against COVID-19 [110].

Ritonavir was discovered in 1992, and it was approved and commercialized in 1996. The classical synthetic route features a convergent coupling of three key intermediates: BOC-core-succinate, 5-wing, and 2,4-wing (Scheme 45) [115].

Polymorphism-related problems and the use of process harsh conditions have motivated synthetic chemists to improve and/or find alternative and sustainable pathways to obtain ritonavir. Due the biological importance of the thiazole core, it is crucial to focus the attention on performant and sustainable synthetic strategies to produce this heterocyclic moiety. 

Isatin and its derivatives have emerged as potential SARS-CoV-2 main protease inhibitors. Recent studies have demonstrated powerful inhibition by isatin compounds bearing a carboxamide moiety at C-5 and aromatic groups with a nitrogen atom in the isatin ring. These two functional groups tethered to the isatin framework are pivotal for the enhanced bioactivity of the molecule (Figure 7).

Substitution of the carboxamide group with an ester or carboxylic acid coincides with a loss of activity due to hydrogen bonds breaking within the protein active site. Rigid aromatic cycles at the N-1 position favor hydrophobic interactions, which results in better fitting in the pocket and subsequently higher inhibition activity [16,116]. 

In the next section, we report current synthetic methodologies to yield substituted thiazoles and nonaromatic isatins. In the last few years, innovative and valid metal/additive-free approaches have been developed for the synthesis of these heterocycles [117]. In this work, we focus our attention on metal-triggered processes.

### Miscellaneous Metal-Catalyzed Heterocycle Synthesis Approaches

The significance of thiazoles is highlighted by the presence of this heterocyclic scaffold in relevant molecules acting as antiviral to anticancer drugs. The wide use of thiazoles and their derivatives in the pharmaceutical industry necessitates the continued development of selective, step-economical, and sustainable synthetic procedures to build those heterocyclic scaffolds.

In 2020, Cao et al. presented a novel and straightforward strategy for the synthesis of decorated thiazoles starting from thioamides **106**, ynals **107**, and alcohols via a Cu(I)-catalyzed reaction (Scheme 46) [118].

The catalytic reaction proceeds in good yields (e.g., **109 a–d**) with high regioselectivity and a broad reaction scope. A domino process leads to the formation of new C–S, C–N, and C–O bonds to obtain functionalized heterocycles in a one-pot fashion starting from easy-to-handle and cheap starting materials.

Cu catalysts have proven very effective for thiazole synthesis. Jiao and coworkers reported a practical and efficient aerobic oxidative sulfuration/annulation protocol to thiazoles via multiple C(*sp*^3^)–H bond cleavage (Scheme 47) [119].

Simple aldehydes **110**, amines **87**, and elemental sulfur can be employed to construct thiazoles through an oxidative sulfuration/cyclization pathway. This methodology shows elevated tolerance toward various functional groups, thus achieving a large array of heterocyclic frameworks in sufficient to good yields. Cheap Cu(I) salt is used as a precatalyst, and environmentally friendly molecular oxygen is used as an oxidant. The whole process is sustainable and appealable for further scale-up.

Pan’s research group exploited heterogeneous palladium catalysts to promote complex thiazole formation using thiobenzamides and isonitriles as precursors (Scheme 48) [120].

A multistep cascade approach has been proposed. The reaction is triggered by Pd-metalated phosphorus-doped porous organic polymers (POPs). This developed catalytic system was shown to be highly stable, efficient (no loss in catalytic activity after several runs), and easy to recover from the reaction mixture. The scope of reaction encompasses a plethora of isonitriles and benzothiamides (e.g., **113 a–d**), obtaining excellent yields of the desired targets.

Isatin (1*H*-indole-2,3-dione) and its derivatives are found in nature, and their framework is present in a wide range of active compounds as apoptosis-inhibitory, anticonvulsant, antiviral, and antifungal agents. Classical synthetic approaches include the Sandmeyer methodology, Stolle procedure, Martinet synthesis, and Gassman procedure [121]. Generally, traditional synthetic methodologies lead to isatins in good yield, but the use of harsh reaction conditions, the use of highly reactive reagents, and the formation of byproducts decrease their sustainability. The introduction of metal catalytic systems has resulted in mild and environmentally friendly methodologies to achieve isatins.

In 2017, Das and coworkers developed an innovative method for the Cu(I)-catalyzed oxidative amidation of 2-aminophenylacetylenes using air oxygen as a green oxidant (Scheme 49) [122].

This methodology achieves substituted isatins in good yields. The use of inexpensive Cu(I) salt as a catalyst and air as an oxidant makes this synthesis practical and environmentally friendly.

Wu et al. proposed a facile strategy to obtain isatin scaffolds via FeCl_3_-triggered intramolecular Friedel–Crafts alkylation (Scheme 50) [123].

The reaction proceeds through direct intramolecular addition and oxidation in a one-pot manner, yielding substituted isatins in high yield. This cost-effective strategy is based on a stable precursor and simple operative protocol. Atom economy and gram-scale application are accomplished, and the use of an earth abundant iron catalyst is desirable due to its low cost and nontoxicity.

In later work, the same research group showed the capability of RuCl_3_ to promote C(*sp*^2^)-H activation/oxidative acylation to obtain isatin compounds starting from α-hydroxy amides **117** (Scheme 51) [124].

RuCl_3_ activates aromatic hydroxyl amides **107**, promoting their cyclization in mild conditions. The methodology is carried out in mild conditions and shows a high tolerability toward various functional groups tethered to the heterocyclic scaffold (e.g., **115 i–l**). Ruthenium works both as an oxidant and as an activator; thus, a stoichiometric amount of transition metal is required.

## 7. Conclusions

This review presented recent methodologies applied to produce various aromatic and nonaromatic heterocyclic scaffolds that have gained interest in the fight against the ongoing SARS-CoV-2 outbreak. We reported the biological mechanisms of promising old and new APIs showing anti-SARS-CoV-2 activities and the industrial and classical approaches to their synthesis. A comparison between traditional and metal-catalyzed protocols revealed the general improvement in reaction conditions and operability due to metal catalysis. The employment of a catalytic system leads to elevated yields, high target selectivity, and a high purity of the desired scaffolds; furthermore, the use of metal catalysts allows reducing and sometimes avoiding the use of strong and unrecoverable reagents such as mineral acids, strong bases, or necessary additives to promote the formation of the final products. Unfortunately, the majority of metal-catalyzed heterocyclization reactions require high temperatures, stoichiometric oxidants, or sacrificial reagents to restore the metal oxidation state and restart the catalytic cycle. The use of unrecoverable boiling-point solvents is necessary and represents a critical aspect, as it corresponds to a decrease in process sustainability. Recently, ongoing modifications of these catalytic procedures have exploited the use of greener additives such as air or oxygen as natural oxidizing agents or ecofriendly solvents, e.g., alcohols or easily recoverable halogenated solvents. Key features of this review are presented in Figure 8.

Continued research on heterocyclic scaffold synthesis is crucial to face the crisis caused by the pandemic, as well as lead to the development of innovative, practical, and easily scalable processes to produce new drugs or known APIs.

## Data Availability

No new data were created or analyzed in this study. Data sharing is not applicable to this article.

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
