# Peer review of "Metal-Promoted Heterocyclization: A Heterosynthetic Approach to Face a Pandemic Crisis"

_molecules, 2021, doi:10.3390/molecules26092620_

Round 1

Reviewer 1 Report

In this manuscript, the authors review a very hot topic "Metal-promoted Heterocyclization, a heterosynthetic approach to Face a Pandemic Crisis". The authors are experts on this topic and already have some previous articles related to metal-catalyzed heterocycle synthesis.

The review is written in a concise, organized and easy to read and follow manner. In its form, it shows a comparison between traditional and some of the more recent metal-catalyzed protocols. It discusses the advantages and disadvantages of each methodology indicated and points out which are the most suitable from an industrial point of view.

The schemes, tables and figures are fully detailed. The introduction and the reference section are adequate and up-to-date.

Therefore, I think the article is suitable for publication in Molecules, although the final decision depends on the editor.

I have some minor correction:

In line 95: Scheme 1. Aromatic chlorine is missed in compound 1.

In line 208 says “aroylakynes 22 (Scheme 15)” but in line 210 is written scheme 14. From here, the number of rest of the schemes is wrong.

In line 220 say “(Ph3P)AuCl/AgOtf” but should say “(Ph3P)AuCl/AgOTf”

Line 229 ”proparyilamines 25” and should say “propargylamines”.

Line 281 and 290. Scheme 19 is written twice.

In line 288, ref. 65 doesn’t fit with one in the reference list.

Line 318 “conversion (e.g. 45b)”, and should say “conversion (e.g. 45c)”

In line 345 “Xang research group” (ref 70) but should say “Wan research group”

In line 460, “to access 2-substitued benzimidazoles” should say “substituted” and same comment in line 464.

In line 484 “oxidation ws able”.

In line 512, “anilines 89 and dioxzolones 90”, should say dioxazolone

In line 614 says “is demonstrated in Su work.” Reference at Su work (J. Chem. Research, 2009, 27-29) is not listed in the reference section.

References

In references 1, 3 and 4, published year should be written in bold.

In ref.26: it says “Chem. – A Eur. J.” and should say “Chem. Eur. J.”

In ref 43: It says “RuCl2(dmso)4”, DMSO should be in capital letter.

Author Response

All the corrections made are highlighted in yellow.

In line 95: Scheme 1. Aromatic chlorine is missed in compound 1. Added Cl in the molecule scaffold

In line 208 says “aroylakynes 22 (Scheme 15)” but in line 210 is written scheme 14. From here, the number of rest of the schemes is wrong. Scheme numbers updated.

In line 220 say “(Ph3P)AuCl/AgOtf” but should say “(Ph3P)AuCl/AgOTf”Correction Made.

Line 229 ”proparyilamines 25” and should say “propargylamines”. Correction Made.

Line 281 and 290. Scheme 19 is written twice. Scheme number correct and update.

In line 288, ref. 65 doesn’t fit with one in the reference list. Reference corrected.

Line 318 “conversion (e.g. 45b)”, and should say “conversion (e.g. 45c)” Correction made.

In line 345 “Xang research group” (ref 70) but should say “Wan research group” Correction made.

In line 460, “to access 2-substitued benzimidazoles” should say “substituted” and same comment in line 464. Correction made.

In line 484 “oxidation ws able”. Correction made.

In line 512, “anilines 89 and dioxzolones 90”, should say dioxazolone Correction made.

In line 614 says “is demonstrated in Su work.” Reference at Su work (J. Chem. Research, 2009, 27-29) is not listed in the reference section. Reference added and reference section updated.

Reference 1,3,4, 26 and 43 updated.

In attachment the update file with the corrections.

Reviewer 2 Report

The review by Rossi and co-workers represents a nice piece of chemistry devoted to the present pandemic emergency and deserves attention.

The topic reported in the manuscript aims to shine some light on the use of specific synthetic tools for the synthesis of heterocyclic scaffolds to be used for the construction of biologically active compounds.

The subject is of great interest and originality for a synthetic chemist.

This represents a comprehensive view of this subject.

The overall work is correctly organized and written although in some cases the description of the single synthesis aspects seems too short with few details. From scheme 2 on, short comments are given to the single reactions. It would be important to emphasize some aspects of the single approaches to give further details, when relevant.

The conclusions are fine although simple; a general scheme giving a final picture of the methods could be inserted.

The manuscript can be published after minor revisions.

Author Response

Thanks for your suggestions on the work. I add a final figure in order to point out the key subjects of this review and give it a better impact.

In attachment you will find the update version of the work with the corrections highlighted in yellow. 
